# Impact of Pre-Analytical and Analytical Variables Associated with Sample Preparation on Flow Cytometric Stainings Obtained with EuroFlow Panels

**DOI:** 10.3390/cancers14030473

**Published:** 2022-01-18

**Authors:** Łukasz Sędek, Juan Flores-Montero, Alita van der Sluijs, Jan Kulis, Jeroen te Marvelde, Jan Philippé, Sebastian Böttcher, Marieke Bitter, Joana Caetano, Vincent H. J. van der Velden, Edwin Sonneveld, Chiara Buracchi, Ana Helena Santos, Margarida Lima, Tomasz Szczepański, Jacques J. M. van Dongen, Alberto Orfao

**Affiliations:** 1Department of Microbiology and Immunology, Medical University of Silesia in Katowice (SUM), 41-808 Zabrze, Poland; lsedek@sum.edu.pl; 2Cancer Research Center (IBMCC, USAL-CSIC), Department of Medicine and Cytometry Service (NUCLEUS), University of Salamanca (USAL), 37007 Salamanca, Spain; jflores@usal.es (J.F.-M.); j.j.m.van_dongen@lumc.nl (J.J.M.v.D.); 3Institute of Biomedical Research of Salamanca (IBSAL), 37007 Salamanca, Spain; 4Center of Biomedical Network Research in Cancer (CIBER ONC), Carlos III Institute of Health, 28029 Madrid, Spain; 5Department of Immunology, Leiden University Medical Center (LUMC), 2300 RC Leiden, The Netherlands; A.J.van_der_Sluijs-Gelling@lumc.nl; 6Department of Pediatric Hematology and Oncology, Medical University of Silesia in Katowice (SUM), 41-800 Zabrze, Poland; jkulis@sum.edu.pl (J.K.); tszczepanski@sum.edu.pl (T.S.); 7Department of Immunology, Erasmus MC, University Medical Center Rotterdam, 3015 CN Rotterdam, The Netherlands; j.temarvelde@erasmusmc.nl (J.t.M.); v.h.j.vandervelden@erasmusmc.nl (V.H.J.v.d.V.); 8Department of Diagnostic Sciences, Ghent University, 9000 Ghent, Belgium; jan.philippe@ugent.be; 9Special Hematology Laboratory, Medical Clinic III, Hematology, Oncology and Palliative Medicine, Rostock University Medical Center, 18057 Rostock, Germany; Sebastian.Boettcher@med.uni-rostock.de; 10European Scientific Foundation for Laboratory Hemato Oncology (ESLHO), 2333 ZA Leiden, The Netherlands; w.m.bitter@lumc.nl; 11Clinical Flow, Hemato-Oncology Unit, Champalimaud Foundation, 1400-038 Lisboa, Portugal; joanafilipacaetano@gmail.com; 12Princess Maxima Center for Pediatric Oncology, 3584 CS Utrecht, The Netherlands; esonneveld@skion.nl; 13Pediatric Clinic of Milano-Bicocca, Tettamanti Research Center, Monza (TRC), 20900 Monza, Italy; chiara.buracchi@gmail.com; 14Department of Hematology, Central Hospital of Porto (CHP), 4099-001 Porto, Portugal; anahrs@gmail.com (A.H.S.); margaridalima@chporto.min-saude.pt (M.L.)

**Keywords:** flow cytometry, standardization, immunophenotyping, protocol, anticoagulant, sample storage, leukemia, lymphoma, multiple myeloma

## Abstract

**Simple Summary:**

Objective interpretation of flow cytometry may be hampered by a lack of standardized sample preparation procedures. The EuroFlow consortium conducted a series of experiments to determine the potential impact of different pre-analytical and analytical factors on the variability of results in terms of relative cell populations distribution and marker expression levels. The experiments were performed on healthy donors and patients with different hematological malignancies (e.g., acute leukemia, lymphoma, multiple myeloma, and myelodysplastic syndrome) to mimic real-world clinical settings. Overall, the results showed that sample storage conditions, anticoagulant use, and sample processing protocol might need to be tailored for sample and cell type(s), as well as to the specific markers evaluated. However, defining of well-balanced boundaries for storage time to 24 h, staining-acquisition delay to 3 h, and choosing a washing buffer of pH within the range of 7.2 to 7.8 would be a valid recommendation for most applications and circumstances described herein.

**Abstract:**

Objective interpretation of FC results may still be hampered by limited technical standardization. The EuroFlow consortium conducted a series of experiments to determine the impact of different variables on the relative distribution and the median fluorescence intensity (MFI) of markers stained on different cell populations, from both healthy donors and patients’ samples with distinct hematological malignancies. The use of different anticoagulants; the time interval between sample collection, preparation, and acquisition; pH of washing buffers; and the use of cell surface membrane-only (SM) vs. cell surface plus intracytoplasmic (SM+CY) staining protocols, were evaluated. Our results showed that only monocytes were represented at higher percentages in EDTA- vs. heparin-anticoagulated samples. Application of SM or SM+CY protocols resulted in slight differences in the percentage of neutrophils and debris determined only with particular antibody combinations. In turn, storage of samples for 24 h at RT was associated with greater percentage of debris and cell doublets when the plasma cell disorder panel was used. Furthermore, 24 h storage of stained cells at RT was selectively detrimental for MFI levels of CD19 and CD45 on mature B- and T-cells (but not on leukemic blasts, clonal B- and plasma cells, neutrophils, and NK cells). The obtained results showed that the variables evaluated might need to be tailored for sample and cell type(s) as well as to the specific markers compared; however, defining of well-balanced boundaries for storage time, staining-to-acquisition delay, and pH of washing buffer would be a valid recommendation for most applications and circumstances described herein.

## 1. Introduction

Flow cytometry (FC) is a key diagnostic tool in hemato-oncology. However, objective interpretation of FC results may still be hampered by limited technical standardization. In recent years, the EuroFlow consortium has designed, validated, and proposed a large set of screening tubes and disease-oriented panels of monoclonal antibody combinations [1]. In addition, EuroFlow has also proposed standard operating procedures (SOPs) for the flow cytometer setup and calibration, quality assessment, antibody titration, and sample staining [1,2,3,4,5,6,7]. In parallel, an external quality assessment program has been set up for centers that have adopted and that use the EuroFlow panels and SOPs [2,3,8,9,10,11]. Nevertheless, several variables that might have an impact on the pre-analytical and analytical phases of the proposed flow cytometry tests have not been fully addressed by EuroFlow and emerge as potential sources of results variability. Based on the long-term experience of the laboratories gathered within the EuroFlow Consortium, several factors related to sample preparation were identified as a potential source of unwanted levels of variability, which might negatively impact intra- and inter-laboratory reproducibility and comparability of results. In detail, these factors included, among other variables, the use of different anticoagulants for sample collection, distinct time intervals between sample collection, sample preparation and data acquisition, storage temperatures (room temperature (RT) vs. 4 °C), the specific approaches used for staining of cell surface membrane-only (SM) vs. cell surface plus intracytoplasmic (SM+CY) markers, and the pH of buffers used during sample preparation and/or acquisition in the flow cytometer.

Nowadays, different anticoagulants are used in routine diagnostics in hemato-oncology, of which dipotassium (K2) or tripotassium (K3) ethylene diaminetetraacetic acid (EDTA) and heparin are currently recommended for FC immunophenotyping [12,13]. EDTA and heparin prevent triggering the coagulation cascade via different mechanisms. Thus, whereas EDTA binds divalent metal ions like Ca^2+^ [14], heparin enhances the antithrombin inhibitory activity against several coagulation factors [15]. WBC and differential counts, as well as polymerase chain reaction (PCR)-based molecular assays require EDTA-anticoagulated samples [16]. In contrast, heparin is used for conventional cytogenetics and functional cellular assays. Regarding FC immunophenotyping, several studies suggest that heparin best preserves granulocyte antigens, for instance in quantitative assays and maturational studies [17,18], and it has been recommended for analysis of MDS [19]; however, it is not suitable for morphological assessment and causes artefactual increase of CD11b expression on monocytes [17,20]. In turn, for lymphocytes, EDTA provides longer marker expression stability over time [21], while it has also been associated with decreased expression (vs. heparin) of markers that contain Ca^2+^ binding domains in their NH_2_-terminal regions, such as CD11b [22].

In routine diagnostic settings, differences in time intervals between sample collection, staining, and acquisition occur due to logistic reasons (e.g., whether a sample is collected and analyzed locally or centrally and/or the laboratory workload). Aged samples usually contain more cell debris and dead cells which are associated with decreased marker expression levels and higher amounts of non-specific binding of antibodies and/or fluorochromes (e.g., APC-H7 [6,13,23]) that might cause artefactual variations in median fluorescence intensity (MFI) signals. Such variability in MFI levels of individual markers may be further enhanced by the use of different sample preparation procedures (e.g., SM vs. SM+CY) for staining the specific antibody clones/reagents within the same antibody panel, as with the B-cell precursor acute lymphoblastic leukemia (BCP-ALL), T- and NK-cell chronic lymphoproliferative disorder (T-CLPD and NK-CLPD, respectively) panels.

Finally, the MFI values obtained for different fluorochromes might also depend on the buffers used during sample preparation, storage, and acquisition in the flow cytometer. The most popular buffer used for sample preparation for FC is phosphate-buffered saline (PBS) with bovine serum albumin (BSA). However, the concentration of BSA and the pH of the solutions used vary significantly among distinct manufacturers and centers. Different concentrations of hydrogen ions, as well as different protein contents, combined with variable salt formulations of these solutions may have an impact on the kinetics of the antibody-epitope binding and/or fluorochrome emission profile, as extensively described for fluorescein isothiocyanate (FITC) [18,24]. The differences in antibody binding capacity influence the total fluorescence emission, which ultimately translates into differences in MFI values.

Here we report on studies conducted by EuroFlow to assess the impact of several of the above FC sample preparation-associated variables on the MFI of markers included in frequently used EuroFlow antibody combinations and panels in the diagnostics settings, and their impact on the identification and enumeration of different cell populations present in normal and patient peripheral blood (PB) and bone marrow (BM) samples.

## 2. Materials and Methods

### 2.1. General Immunophenotypic Procedures

The following variables were considered: (i) the anticoagulant (K3 EDTA vs. sodium heparin), (ii) age of sample (0 vs. 24 h) and age of staining (0 vs. 3 h), (iii) the sample staining procedure (staining for SM markers only vs. SM+CY markers), and (iv) the pH and protein contents of the washing buffer. The impact of the abovementioned variables on the staining intensity of immunophenotypic markers and the relative distribution of normal and pathological cells in PB and BM samples was evaluated based on separate sets of experiments (for antibody details, see Appendix A). Unless otherwise specified, PB and BM samples were collected in EDTA and/or heparin and prepared at each participating site following the EuroFlow SOPs for staining of SM markers only and/or SM+CY markers and for instrument set up, calibration and data acquisition in FACS Canto II flow cytometers (Becton Dickinson Biosciences (BD), San José, CA, USA), as previously described [2]. Briefly, incubation time for staining of cell SM markers was 30 min, while fixation and permeabilization time for CY marker staining with Fix & Perm (Life Technologies, Carlsbad, CA, USA) was 15 min. In turn, for cell SM-only stainings, non-nucleated red cells were lysed with FACS Lysing solution diluted 1:10 (*v/v*) in distilled water (BD). For data analysis, Infinicyt software (Cytognos SL, Salamanca, Spain) was used. For each data file, MFI values per marker were recorded for all identifiable cell populations, together with the percent distribution of debris, cell doublets, and each relevant cell population identified in the sample. MFI differences between directly compared conditions were calculated as [(MFI_A_ − MFI_B_)/MFI_A_] × 100%, where MFI_A_ corresponds to MFI values in ‘condition A’ and MFI_B_ to the MFI values of altered ‘condition B’. MFI differences within the ±30% tolerance range were considered as similar, whereas the differences below 70% and above 130% were averaged separately and used to calculate the relative percent decrease and increase in MFI, respectively. For calculation of mean percentage of individual cell populations, debris and doublets, ratio values exceeding a 10-fold difference were considered as outliers and excluded from further analysis.

### 2.2. Patient and Normal Samples and Staining Antibody Combinations per Set of Experiments

A total of 3 BM samples from healthy donors and 2 BM samples of patients with myelodysplastic syndromes collected in parallel in K3-EDTA and in sodium heparin, were used to evaluate the impact of different anticoagulants on (i) the staining profiles of markers included in tube 1 of the EuroFlow AML/MDS antibody panel, and (ii) the overall distribution of normal and abnormal cell populations (Appendix A). To eliminate the possible influence of time, all samples were processed within 12 h after collection at room temperature (RT). For all experiments investigating the impact of age of sample and age of staining on immunophenotypic results and cell distribution, PB and BM samples collected in K3-EDTA from B-cell chronic lymphoproliferative disorder (B-CLPD; *n* = 19), multiple myeloma (MM; *n* = 5) and monoclonal gammopathy of undetermined significance (MGUS; *n* = 5) patients, were used. In the experiments devoted to investigating the impact of age of sample only, 7 BM samples from patients diagnosed with ALL were collected and stained with the EuroFlow Acute Leukemia Orientation Tube (ALOT) tube at the same center (*n* = 2 centers) in duplicate: immediately upon collection and after storage for 24 h at RT. In a different set of experiments, the impact of the age of sample (0 h vs. 24 h) kept at RT was evaluated in parallel to the age of staining (0 h vs. 3 h kept at 4 °C) using BM and PB samples stained with the OneFlow lymphoid screening tube (LST; *n* = 13 samples), plasma cell disorder (PCD; *n* = 10 samples), and B-CLPD-tube 1 (*n* = 6 samples) reagent kits (BD). For experiments addressing the influence of different staining protocols, a total of 23 EDTA-anticoagulated BM samples from BCP-ALL (*n* = 5), T-CLPD (*n* = 8), NK-CLPD (*n* = 5) patients and normal EDTA-anticoagulated BM (*n* = 5) and PB (*n* = 3) samples were used. The percent distribution of all cell populations and the MFI of backbone markers of the three mentioned EuroFlow antibody panels were compared between the protocol applied for staining of SM-only and SM+CY tubes on both the normal and aberrant cell populations identified in each sample (CD19, CD34, and CD45 for the BCP-ALL panel; CD4, CD8, smCD3, and CD45 for the T-CLPD panel; CD56 and CD45 for the NK-CLPD panel; Appendix A). The BCP-ALL panel was applied to normal and patient BM samples, whereas the T- and NK-CLPD panels were used to stain patient BM and normal PB samples. Finally, to assess the influence of different pH and protein contents of the washing buffer on MFI levels of individual markers and the relative distribution (percent values) of different cell populations, 5 normal EDTA-anticoagulated PB samples collected at 4 different centers (*n* = 20) were treated during the staining procedure with 8 different washing buffer conditions: PBS supplemented with 0.09% of sodium azide (Sigma-Aldrich, St. Louis, MO, USA) and 0.2% or 0.5% of BSA (Sigma-Aldrich), adjusted to four different final pH (7.2 vs. 7.4 vs. 7.6 vs. 7.8). All those latter samples were stained with the LST combination in four EuroFlow laboratories.

The study was conducted in accordance with the Declaration of Helsinki, and approved by the Ethics Committee of the Medical University of Silesia in Katowice (KNW/0022/KB1/153/I/16/17 approved on 3 October 2017), Ethics Committee of the Ghent University Hospital (EC 2016/1138, B670201629681, approved on 29 November 2016).

### 2.3. Statistical Methods

To determine the statistical significance of differences observed in the MFI values of individual markers and percent values of specific cell populations between groups of ≥5 samples, the (paired or unpaired) Student’s *t*-test and the Wilcoxon signed rank test were used for parametric and non-parametric data (as assessed by the Shapiro–Wilk test), respectively. In turn, either the ANOVA or the Kruskal–Wallis tests were used for comparisons among three or more groups. *p*-values < 0.05 were used as cutoff for statistical significance. All statistical analyses were performed with the Statistica 13 software (Tibco, Palo Alto, Santa Clara, CA, USA) and graphics shown in Figures 1–5 were generated with Prism software (GraphPad, San Diego, CA, USA).

## 3. Results

### 3.1. Impact of the Anticoagulant on Cell Distribution and Staining Profiles

The use of sodium heparin vs. K3-EDTA as anticoagulants had no impact on the relative distribution of debris, cell doublets and both normal and pathological nucleated cells neither in the normal (*n* = 3) nor in the MDS BM samples analyzed (*n* = 2), except for monocytes which were found to be represented at higher percentages in EDTA-anticoagulated vs. heparinized sample median (range): 3.1% (2.8–3.7%) vs. 2.1% (1.0–2.6%), respectively; *p* = 0.02 (Figure 1A). In general, antibody-associated MFI levels were similar for all eight markers included in tube 1 of the EuroFlow AML/MDS panel for all cell populations evaluated in parallel in paired normal and MDS BM samples (Figure 1B). A tendency toward slightly higher MFI values in EDTA- vs. heparin-anticoagulated samples was observed for most markers investigated, however these differences remained within the ±30% tolerance range. Among all markers, CD11b was the only marker associated with the greater mean MFI differences on neutrophils (+254%) and monocytes (+273%) in the majority of heparinized vs. EDTA-anticoagulated BM samples, but these differences did not reach statistical significance (Figure 1B).

### 3.2. Age of Sample

Sequential staining of the same BCP-ALL BM patient samples immediately after collection vs. after storage for 24 h at RT in the same center resulted in similar distribution of leukemic and normal cells in the sample (Figure 2A,B and Appendix A). Additionally, sample storage for 24 h did not influence the percentage of debris and cell doublets.

Assessment of MFI levels revealed that 24 h storage of BM samples from BCP-ALL patients at RT was associated with similar (*p* > 0.05) expression levels of most markers included in the EuroFlow ALOT (Figure 2B). However, selective reduction of MFI values for specific markers in specific subsets of normal and/or leukemia cell populations was observed. Thus, the expression levels of CD45 were significantly (~50%) decreased after 24 h storage at RT (vs. 0 h) on normal mature B- (*p* = 0.03) and T-cells (*p* = 0.02) of most samples, while no significant changes in CD45 expression levels were observed on leukemic blasts, neutrophils, and NK cells. In addition, CD19 expression levels on leukemic blasts were decreased by ~60% in around 60% of samples stored for 24 h (*p* = 0.03 vs. 0 h), with a decreasing trend visible also on normal mature B-cells which however did not reach statistical significance (Figure 2B).

The effect of age of sample was also assessed with the BD OneFlow LST, PCD, and B-CLPD tube 1 reagent kits (see Appendix A for protocol details). Preparation of the samples at 0 h vs. after 24 h storage at RT showed no impact on the distribution of normal and pathological BM and PB cells (and their subsets) neither in the normal nor in the patient samples analyzed (Figure 3A–C and Appendix A). However, higher percentages of debris and cell doublets with the PCD kit were found at 24 h storage vs. 0 h—median (range) of 10.9% (5.5–14.9%) vs. 6.9% (2.9–11.5%), respectively (*p* = 0.01); and 6.6% (5.7–11.1%) vs. 4.5% (3.2–6.5%), respectively (*p* = 0.01)—in the majority of samples (Figure 3C and Appendix A). No significant differences in MFI levels between time 0 h vs. 24 h were detected for any marker evaluated on any cell population with the use of any of the OneFlow reagent kits (Figure 3D–F).

### 3.3. Age of Staining

Acquisition of samples stained immediately after preparation had been completed vs. after 3 h storage at 4 °C showed no impact on the distribution of normal and pathological BM and PB cells (and their subsets) neither for the normal nor for the patient samples analyzed, regardless of whether samples had been stained the same day they had been collected or after storage for 24 h at RT (Figure 3A–C and Appendix A).

Similarly, no significant differences in the MFI levels of individual markers were observed between samples that were immediately acquired in the flow cytometer vs. those stored for 3 h at 4 °C for any cell population stained with any of the OneFlow reagent kits, regardless of whether samples had been stained fresh or stored at RT for 24 h prior to staining (Figure 3D–F).

### 3.4. Cell Surface-Only (SM) vs. Cell Surface plus Intracytoplasmic (SM+CY) Staining Procedures

For the great majority of normal and pathological nucleated BM and PB cell populations and their subsets, as well as for cell doublets, application of SM and SM+CY sample preparation and staining protocols did not reveal significant differences in their relative distribution (Figure 4A–C). However, the percentage of neutrophils after staining with the T-CLPD panel (but neither the BCP-ALL nor the NK-CLPD panels) was significantly higher for the SM vs. SM+CY conditions -median (range) of 31.3% (6.3–63.5%) vs. 21.5% (4.6–55.0%), respectively (*p* = 0.002) (Figure 4B). In addition, a trend toward higher percentages of clonal CD8^+^ T-cells (*n* = 2) and abnormal CD56^+^ NK-cells (*n* = 3) was also observed.

In contrast, the percentage of debris was significantly higher in the SM+CY vs. SM conditions for samples stained with both the T-CLPD and NK-CLPD panels—median (range) of 27.7% (8.3–44.0%) vs. 9.2% (4.4–20.9%) (*p* = 0.004); and of 26.3% (9.0–47.4%) vs. 9.4% (4.0–35.7%), respectively (*p* = 0.02) (Figure 4B,C). Of note, similar MFI expression levels were observed with both SOPs for all backbone markers stained with the EuroFlow antibody panels evaluated (BCP-ALL, T-CLPD, and NK-CLPD) for normal and pathological BM and PB cells and their subsets (Figure 4D–F).

### 3.5. Washing Buffer (PBS) pH and Protein Contents

A similar distribution of debris, cell doublets and nucleated normal cells, together with similar antigen expression levels were observed between PB samples washed with PBS containing different concentrations of BSA (0.2% vs. 0.5% *w/v*) at identical pH of the washing buffer (Appendix A). Thus, the actual influence of pH was evaluated on averaged cell percentages and MFI values obtained for both BSA concentrations (Figure 5A,B). Overall, no significant differences in the percentage distribution of normal nucleated cells, debris, and cell doublets were observed for all different conditions evaluated (Figure 5A).

In addition, MFI expression levels for the great majority of the 12 LST markers evaluated was stable at different pH conditions of the PBS washing solution. Despite this, a non-significant trend toward higher MFI levels at greater pH values was observed for the two markers that were conjugated with FITC (Igλ and CD8; Figure 5B; Appendix A).

## 4. Discussion

In this study, the impact of different samples, sample collection, and preparation conditions used for FC immunophenotyping were evaluated based on a large set of PB and BM samples collected from both healthy individuals and patients with various hematological malignancies. For each condition compared, the percentage of normal and pathological cells (and their subpopulations), debris, cell doublets, and the staining intensity (MFI) obtained for individual markers contained in different EuroFlow leukemia/lymphoma antibody panels were assessed. Specifically, we focused on potential differences in these parameters that might be associated with the use of different anticoagulants, the time elapsed between sample collection and staining and between sample staining and acquisition in the flow cytometer, the use of different staining protocols (SM vs. SM+CY), and the pH of the washing solutions. The use of sample fixatives and their influence on marker stability was beyond the scope of our study, as it has been extensively investigated and reviewed elsewhere [25,26].

Careful review of guideline papers on technical aspects related to FC immunophenotyping, such as those including recommendations on sample preparation protocols, anticoagulants, optimal sample staining, and storage time and temperature [12,13,16,24,27,28] revealed that universal recommendations that are valid for virtually every sample and cell type and/or even every marker to be evaluated on a variety of cell types, can hardly be found. Therefore, it is important to be aware of—and determine the potential impact of—these variables in specific settings, particularly in those related to national FC-based diagnostic networks, large scale multicenter studies and international clinical trials. To assure optimal levels of standardization appropriate for handling biological specimens, standardized sample preparation solutions and protocols are a prerequisite.

Even though EDTA and heparin are the most widely used anticoagulants recommended for FC leukemia/lymphoma phenotyping, virtually no studies exist in which both anticoagulants had been directly compared in normal and patient samples. Here, we compared the FC staining profiles of paired EDTA- and heparin-anticoagulated BM samples stained freshly (<12 h after collection) to exclude an additive influence of sample storage. For this purpose, normal and MDS BM samples were used. At present, several consensus guidelines provide recommendations on the type of anticoagulant and sample storage time and temperature to be used for optimal FC immunophenotyping [12,16,17,18,21]. Thus, for MDS BM heparin is usually recommended [19] since artefactually decreased CD11b expression levels on monocytes are likely to occur with EDTA [17,20], in line with the non-significant decreasing tendency also found here for CD11b expression levels on both monocytes and neutrophils. These variable differences in CD11b MFI might be due to the fact that anti-CD11b reagents bind to Ca^2+^-dependent conformational epitopes in the CD11b molecule. As reported by Repo et al. [29], this might be corrected by the use of higher amounts of the CD11b antibody (clone D12, i.e., the same as the one tested in the current study) for staining EDTA-anticoagulated samples—e.g., MDS patient samples. The opposite anticoagulant choice (i.e., EDTA) should however be made for staining for the CD138 antigen, whose expression levels on PC were higher in EDTA- vs. heparin-anticoagulated samples [30]. In line with these observations, other authors confirmed that the effect of different anticoagulants might vary depending on the antigen and specimen type being analyzed and that the expression levels of markers such as CD4, CD13, CD33, CD34, CD38, CD45, CD71, and CD117 vary on different cell types between EDTA- and heparin-anticoagulated samples [12,31]. Although the combined influence of storage time and type of anticoagulant was not specifically assessed here, the slight differences in performance of heparin and EDTA as regards the expression levels of markers evaluated in stained and stored PB samples have been described elsewhere [25,31].

One of the most relevant and difficult to standardize FC immunophenotyping variables is the time elapsed between sample collection, staining, and acquisition in the flow cytometer because several different variables might be involved. These include the temperature (RT, ambient temperature, refrigeration at 4 °C) or the need for potential transportation of the sample vs. local storage. In routine practice, the differences in time may result from logistic reasons (need to send samples to a reference laboratory), transportation time and unintentional delays (e.g., weekends or occasional holidays), and/or even the variable laboratory daily workload. Although the upper limit for unfixed specimen storage time has not been previously defined precisely, it is well known that the maximum acceptable specimen age varies as a consequence of the anticoagulant, storage conditions (time and temperature), sample preparation procedure, and type of cells of interest [16]. Refrigerating the samples at 4 °C may retain the original MFI values of individual cell surface antigens, particularly of those expressed by monocytes, neutrophils, and myeloid progenitor cells for periods of 48-h storage or longer [32]. In addition, 4 °C storage of fresh PB and BM samples might even provide acceptable results on expression levels (MFI values) of individual myeloid and lymphoid markers even up to 7–10 days [33]. In this regard, refrigeration of PB samples at 4 °C has also been shown to better stabilize granulocyte and monocyte numbers for up to 72 h, compared to storage at RT [25]. Despite this, it has also been reported that 4 °C refrigeration of PB samples may be selectively deleterious for some lymphocyte subsets such as CD4^+^ T-cells, and storage for >72 h is therefore not indicated for lymphocyte subset analysis [16]. In turn, it has also been suggested that assessment of PB or BM samples of patients with lymphoproliferative disorders after storage provided similar results in terms of expression levels of commonly evaluated lymphoid markers for up to 72 h at 4 °C vs. RT [13]. In contrast, maturation patterns for myelo-monocytic markers in both normal and MDS cells appear not to be consistently retained in samples stored at RT for >48 h after sample collection [13,19].

In the present study, we investigated the impact of aging of BM and PB samples stored at RT on the relative distribution and marker expression levels of multiple normal and abnormal cell populations, as identified with the EuroFlow ALOT, BCP-ALL, LST, B-CLPD-tube 1, and PCD antibody combinations. Overall, no significant effect of the sample aging was observed on the distribution of normal and abnormal populations of nucleated cells for up to 24 h of storage. Only the percentage of debris and cell doublets was significantly higher in stored vs. fresh samples, but only stained with the PCD antibody combination (not ALOT, LST, or the B-CLPD-tube 1).

Twenty-four hours of storage of PB and BM samples at RT was associated with selectively decreased levels of markers expressed on leukemic blasts and normal residual lymphocytes from leukemic BM samples stained with ALOT, but not LST, B-CLPD-tube 1, or PCD. Thus, our results showed that even limited storage of leukemic samples for 24 h at RT may have a detrimental effect on the expression levels of markers like CD19 on ALL blasts (but not on mature B-cells) and CD45 on mature B- and T-cells (but not on leukemic blasts, clonal B- and plasma cells, neutrophils, and NK cells). In fact, the great majority of markers evaluated was not susceptible to the effect of sample aging within 24 h. Some minor differences (within the acceptable ±30% MFI range) were indeed seen, in line with previous observations by Diks et al. who noted that 24-h PB sample storage resulted in an acceptable >10% decrease in marker MFI irrespective of storage temperature [25]. It should be noted that some markers (e.g., CD19 and CD45) behaved differently under the same conditions on distinct cell populations. In this regard, differences were observed between normal and leukemic cells which is generally in line with previous observations by Davis et al., who showed that normal vs. patient samples might behave differently under similar storage conditions [34]. These observations may also be extrapolated to FC MRD assessment, where the use of up to 24-h old BM samples is common practice in reference laboratories performing country-scale centralized MRD analyses. Although successful validation studies designed in these settings—e.g., BCP-ALL [35] and MM [36] EuroFlow MRD panels and markers have been performed with high concordance rates and similarly high sensitivity levels compared with other MRD techniques (e.g., NGS-based techniques)—it is worth pointing out that decreased expression levels of some MRD markers in aged samples might exist. Thus, Soh et al. reported already decreased surface density of CD138 on PC in samples stored for 24 h, with such a decrease being even significantly more prominent in samples stored for 48 h and 72 h, at both 25 °C and at 4 °C. The same authors also noticed that the expression levels of other PC-associated markers like CD229 increased after storage at 25 °C for 72 h (Appendix A) [37]. According to previously published guidelines, acceptable storage time for unfixed PB or BM samples, freshly stored at RT or 4 °C for further diagnostic FC immunophenotyping varies between 12 and 72 h, depending on the specific markers investigated and diagnostic applications [13,18,34,38]. Altogether, these results support the notion that sample quality is not an absolute term and that it should rather be assessed depending on the individual markers tested and their stability (measured as MFI) under different storage times and/or temperature conditions [12].

Interestingly however, storage of stained samples for 3 h did not negatively influence MFI values of a large number of markers evaluated on patient BM and PB samples, neither in terms of cell percentages nor when marker expression levels on abnormal and normal cells were considered (no age of the staining effect). These results fully support previous observations by Diks et al. showing that delayed acquisition of stained PB samples for even 72 h did not have a significant impact on the MFI values of most markers evaluated in normal PB samples except for CD16+CD56 and TCRγδ. The authors concluded that decreased MFI was more prominent in samples stored unstained vs. those stained beforehand, as a result of nonspecific antibody binding to dying cells, and shedding or internalization of surface markers in stored raw samples [25].

As a final additional goal of our study, we evaluated the impact of different pH and protein contents in the washing buffer and sample preparation/staining protocols on the cell distribution and marker expression profiles as assessed by FC. At present, scarce information is available in the literature in this regard. Overall, our results unprecedentedly demonstrated that the use of different staining protocols (cell surface-only vs. cell surface plus intracellular) for individual antibody panels does not have a significant impact on the intensity of expression of the common backbone markers (i.e., CD45, CD34, CD19, CD3, CD4, CD8, and CD56) neither on normal nor on pathological BM and PB cells. In turn, the use of cell surface-only sample preparation protocol resulted in higher percentages of neutrophils for one of three evaluated antibody combinations (T-CLPD), whereas the opposite was true for the percentage of debris determined with the T- and NK-CLPD antibody panels. The latter feature can be supported with an observation that protocols utilizing cell permeabilization usually cause an increase in cell size (i.e., forward light scatter), with the need for more centrifugation steps, and thereby cell death and spin down of cellular debris.

Finally, our results also showed that the staining intensity observed for the great majority of the markers within the EuroFlow LST combination evaluated on normal PB samples does not significantly depend on the pH and protein contents of the washing solution. The entire pH range (i.e., 7.2–7.8) and BSA concentration (0.2% vs. 0.5%) evaluated was equally suitable for determining reliable expression levels of different markers on PB cell populations. However, for some fluorochromes, such as fluorescein isothiocyanate (FITC) which have been reported to be pH-sensitive, pH values of 7.3 or higher might be preferred [18], since the MFI values for the two FITC-conjugated Igλ and CD8 antibodies tested, in fact exhibited a tendency toward increased MFI values at higher pH.

One of the constraints of this work is the relatively limited sample size for each subtype of blood cancer, particularly for some of the evaluations performed. Nonetheless, the strictly controlled conditions of all the experiments performed in parallel under different pre-analytic and analytic sample-associated conditions contribute to the robustness of the results obtained.

## 5. Conclusions

Here we investigated the impact of several sample preparation-associated variables on the flow cytometry immunophenotyping profiles of normal, leukemia, and lymphoma cells. Overall, our results confirm that both heparin and EDTA can be reliably used as anticoagulants for evaluation of lymphocytic, monocytic, and neutrophilic markers in fresh BM of both patient (i.e., MDS) and healthy donors. The MFI values for some markers after 24-h storage of BM and PB samples at RT decrease to a variable extent, depending on specific cell types (e.g., normal vs. leukemic), once the percentages of debris and both normal and pathological cells are not significantly influenced, which confirms the utility of stored clinical samples for diagnostic purposes. Three-hour storage of stained BM and PB patient samples does not affect the MFI levels, regardless of whether the sample is fresh, or it had been stored for 24 h at RT prior to staining. Under these conditions, no differences in the overall distribution of different populations of nucleated cells, cell doublets, and debris were noted. Similarly, different pH (range: 7.2 to 7.8) and protein contents (BSA concentration 0.2% vs. 0.5%) did not clearly influence the relative distribution of different cell populations and their MFI profiles for the different markers evaluated in PB, regardless of the fluorochrome to which a specific antibody was conjugated (lower BSA concentration might be preferred due to economic reasons). Finally, the use of SM and SM+CY staining protocols does not have an impact on the expression profile of the great majority of markers and it does not influence the relative percentage distribution of the different populations of nucleated cell types present in PB and BM samples. However, the percentage of debris might be higher with the SM+CY (vs. SM) staining protocol as a result of cell permeabilization and death caused by an increased number of washing steps.

Altogether, these results suggest that sample storage conditions might need to be tailored for sample and cell type(s) as well as to the specific markers evaluated; however, defining well-balanced boundaries for storage time to 24 h, staining-acquisition delay to 3 h and choosing a washing buffer of pH within the range of 7.2 to 7.8, would be a valid recommendation for most applications and circumstances described herein. For the above reasons, the authors do not recommend any general changes to any of the EuroFlow SOPs used here.

## Figures and Tables

**Figure 1 cancers-14-00473-f001:**
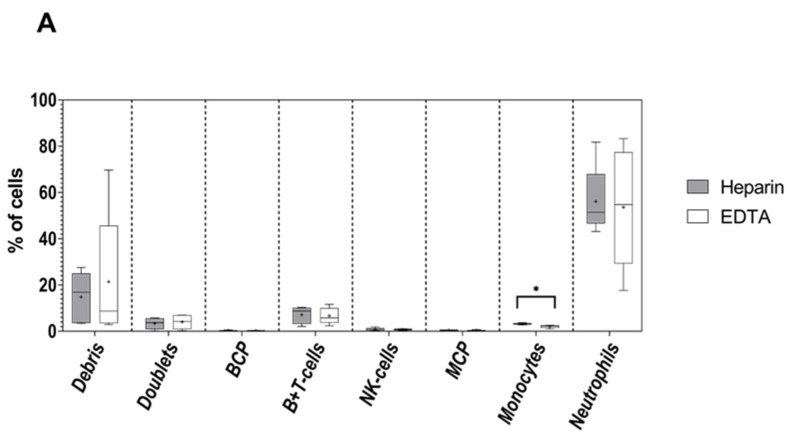
Comparison between the relative distribution of distinct cell populations (**A**) and the corresponding MFI expression levels of individual markers of the AML/MDS tube 1 antibody combination (**B**) evaluated on identifiable cell populations in normal (*n* = 3) and MDS patient BM (*n* = 2) samples in heparin (dark boxes) and EDTA (light boxes) as anticoagulants. The median and mean MFI values are indicated inside the boxes with a line and “+” sign, respectively. The boxes spread between first and third quartile of the measured values and whiskers span between minimal and maximal values. Asterisks indicate significant differences between both anticoagulants (*p* < 0.05). BCP—B-cell precursor cells; MCP—myeloid precursor cells.

**Figure 2 cancers-14-00473-f002:**
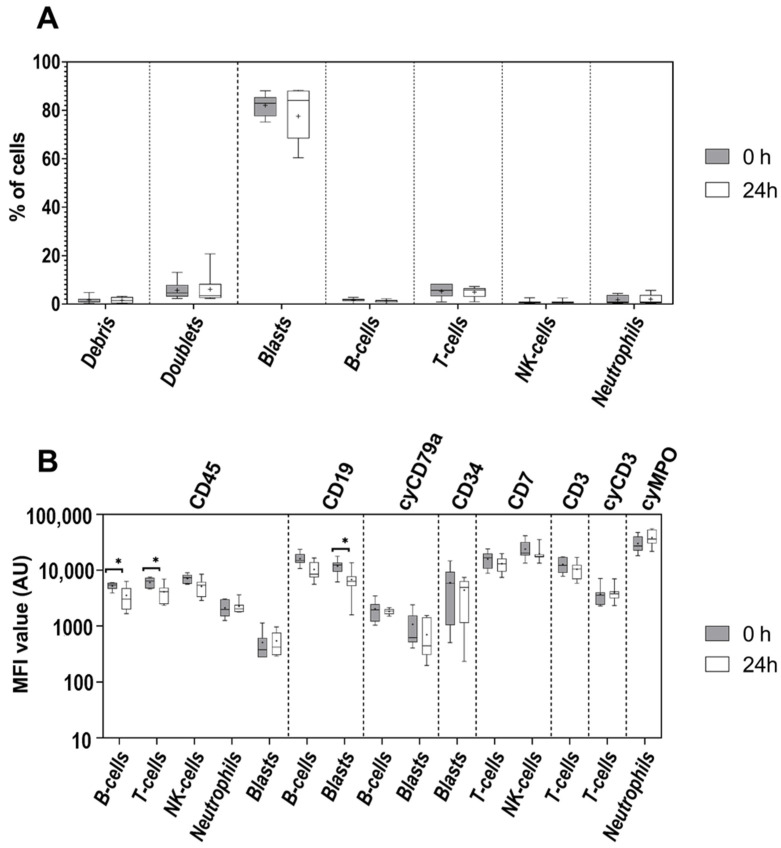
Comparison of the relative distribution of leukemic blasts of BCP-ALL patients (*n* = 7) and normal cell populations identifiable with the ALOT antibody combination (**A**) and MFI expression levels of the corresponding markers (**B**) evaluated on normal and leukemic cell populations from BCP-ALL patients. Dark boxes represent samples acquired fresh, and light boxes represent those acquired after 24-h storage at RT. The median and mean MFI values are indicated inside the boxes with a line and “+” sign, respectively. The boxes spread between first and third quartiles of the obtained MFI values and whiskers span between minimal and maximal values. Asterisks indicate significant differences between both time points (*p* < 0.05).

**Figure 3 cancers-14-00473-f003:**
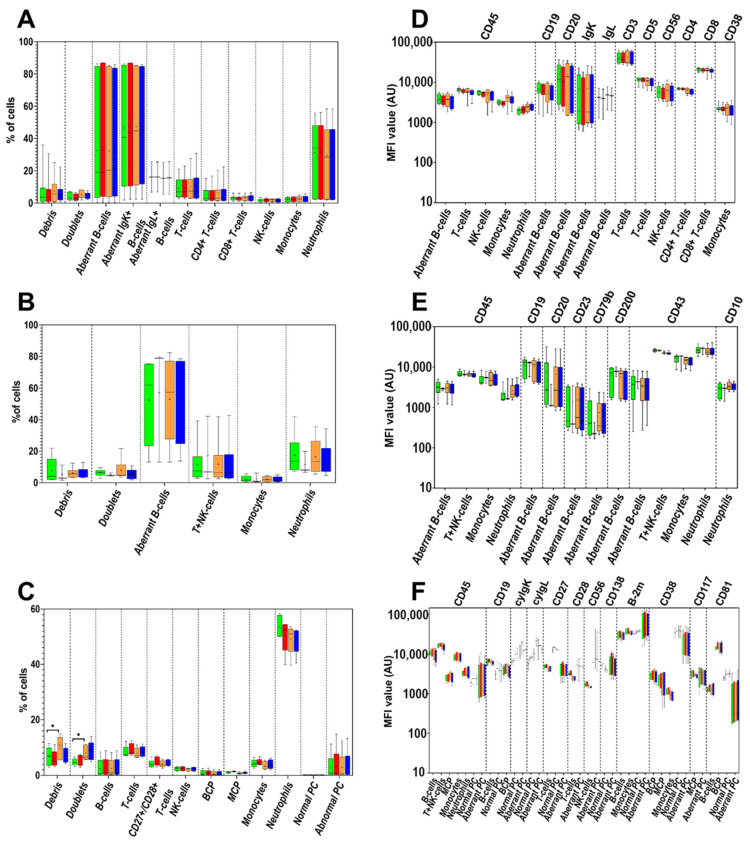
Comparison of the relative distribution of aberrant (clonal) B- and plasma cells and normal cell populations identifiable with the LST (*n* = 13; **A**), B-CLPD tube 1 (*n* = 6; **B**) and PCD antibody combination (*n* = 10; **C**) and the MFI values of individual markers expressed by aberrant and normal cells stained with LST (**D**), B-CLPD tube 1 (**E**) and the PCD antibody combination (**F**). The samples were acquired at different time points: within 15 min after staining (green boxes), after 3-h storage at 4 °C (red boxes), stored overnight at RT, stained at +24 h from collection and acquired within 15 min after staining (orange boxes) or after 3-h storage at 4 °C (dark blue boxes). The median and mean MFI values are indicated inside the boxes with a line and “+” sign, respectively. The boxes spread between first and third quartile of the obtained MFI values and whiskers span between minimal and maximal values. Asterisks indicate significant differences between particular time points (*p* < 0.05). BCP—B-cell precursor cells; MCP—myeloid precursor cells; PC—plasma cells.

**Figure 4 cancers-14-00473-f004:**
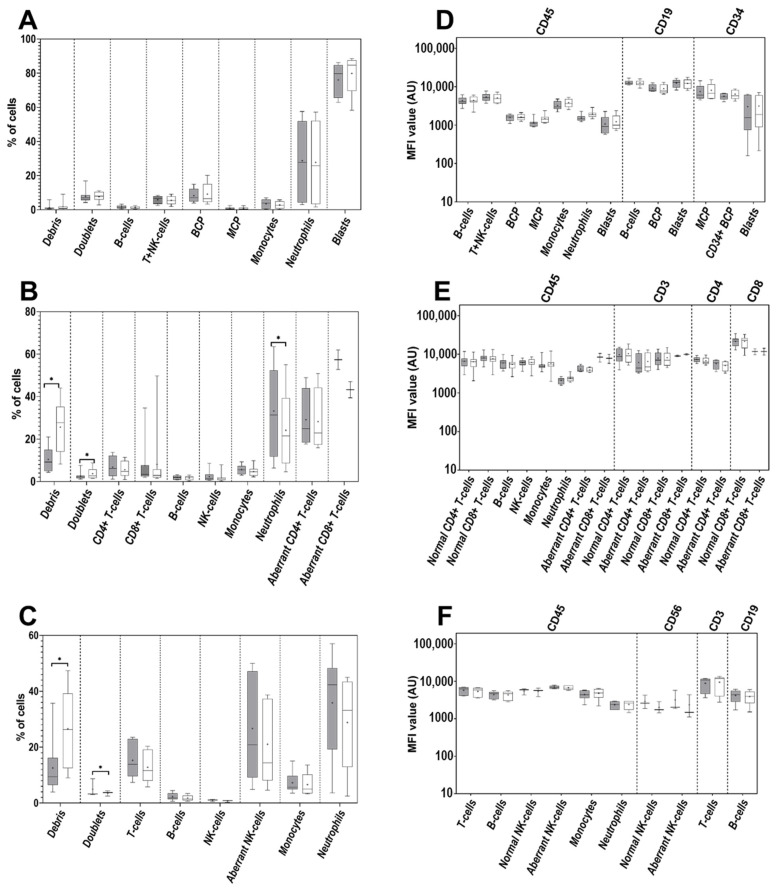
Comparison of the cell surface-only vs. cell surface plus intracellular staining protocols; panels (**A**–**C**): relative distribution of the normal and aberrant cell populations identifiable with the BCP-ALL (*n* = 5), T-CLPD (*n* = 8) and NK-CLPD (*n* = 5) antibody combinations, respectively; panels (**D**–**F**): backbone markers MFI on normal and aberrant cell populations stained with BCP-ALL, T-CLPD and NK-CLPD panels. Dark boxes represent cell surface only- (SM) and light boxes cell surface plus intracellular (SM+CY) staining protocol. The median and mean MFI values are indicated inside the boxes with a line and “+” sign, respectively. The boxes spread between first and third quartile of the obtained MFI values and whiskers span between minimal and maximal values. Asterisks indicate significant differences in MFI between particular time points (*p* < 0.05). BCP—B-cell precursor cells; MCP—myeloid precursor cells; PC—plasma cells.

**Figure 5 cancers-14-00473-f005:**
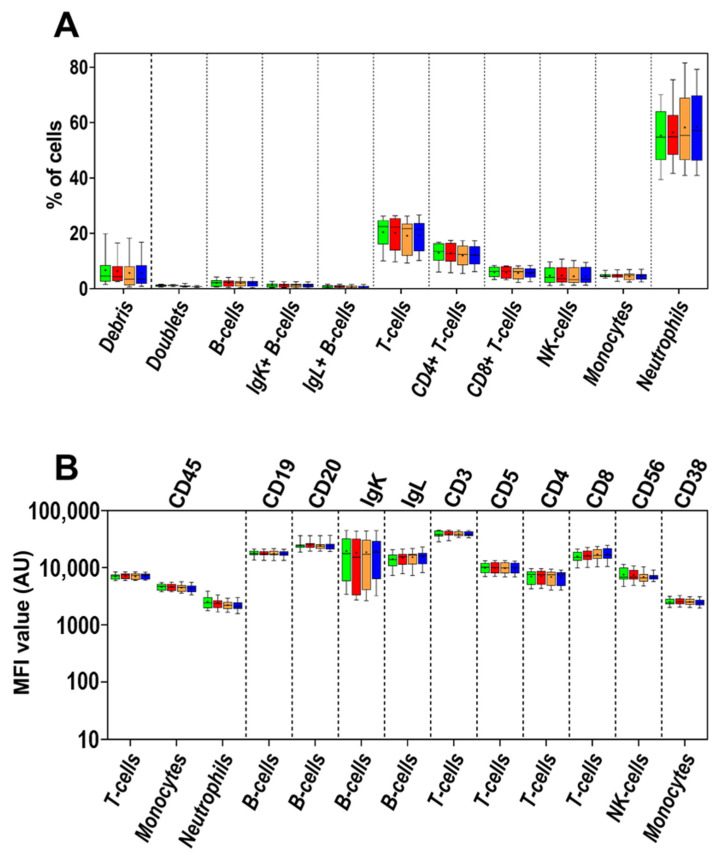
Impact of different pH of the washing buffer on the relative distribution (**A**) and expression levels (**B**) of individual LST markers as assessed on different cell populations in normal PB samples (*n* = 20). The plots represent percentages and MFI levels at different pH values, averaged for both BSA concentrations; red boxes—pH = 7.2; green boxes—pH = 7.4; orange boxes—pH = 7.6; dark blue boxes—pH = 7.8. The median and mean MFI values are indicated inside the boxes with a line and “+” sign, respectively. The boxes spread between first and third quartile of the obtained MFI values and whiskers span between minimal and maximal values. Asterisks indicate significant differences between particular time points (*p* < 0.05).

## Data Availability

For original data, please contact the corresponding author orfao@usal.es.

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
