# Peer review of "Impact of Pre-Analytical and Analytical Variables Associated with Sample Preparation on Flow Cytometric Stainings Obtained with EuroFlow Panels"

_cancers, 2022, doi:10.3390/cancers14030473_

Round 1
Reviewer 1 Report
The authors present a well written follow-up on EuroFlow SOP regarding unaddressed pre-analytic and analytic variables that may impact the final results. While many of these variables have been addressed by others in general flow cytometry, this manuscript show their relevance in the EuroFlow SOP.
This reviewer should have like to see 2 additional information that would enhance this manuscript but can accept that the authors may not have this information.
- Study of samples stored at 4 C for 48-72 hrs because of its impact on transport and workflow.
- Discussion of any impact on diagnosis, particularly on MRD % on BCP-ALL patients when sample is stored for 24 hours at RT due to altered MFI of some antigens.
Reviewer 2 Report
In this work, Sędek et al. investigated the impact of several sample preparation-associated variables on the flow cytometry immunophenotyping profiles of normal, leukemia and lymphoma cells. They showed that higher percentages of monocytes in EDTA- vs heparin-anticoagulated samples, slight differences of the percentage of neutrophils and debris in SM or SM+CY protocols with T-CLPD and NK-CLPD antibody combination, and greater percentage of debris and cell doublets in plasma cell disorder panel and lower MFI levels of CD19 and CD45 on mature B- and T-cells in ALOT panel for storage of samples for 24h vs oh at RT.
1. The strength of the paper is comparing flow cytometry immunophenotyping profiles under identical conditions excepting the sample preparation-associated variables, which lends credibility to their conclusions. However, the majority of their points addressed in this study have been previously studied. For lymphoma and leukemia, the sample size is relatively small for each subtype of blood cancer, it would be more helpful to have the data from an increased number of samples with different specific subtypes.
2. How did the authors decide which variables and the range of selected items to be analyzed? Authors can describe these in the introduction sections.
3. There are many markers in EuroFlow panels. The authors can give a summary of the influences of the sample preparation-associated variables on the other markers that aren’t mentioned in Text. For example, CD138 has been suggested to be preferred to be stored in EDTA due to the decreasing of signal in heparin (Soh KT et al., 2021) and an aged samples changed antigen expression of CD16 (van der Velden, V.H.J. et al. 2021).
4. Authors may provide a list of modifying items for EuroFlow SOP according to the results in the paper. It would be helpful for users.
5. The resolution of Figure 3 and Figure 4 needs to be improved.
